# Standardization of a Preclinical Colon Cancer Model in Male and Female BALB/c Mice: Macroscopic and Microscopic Characterization from Pre-Neoplastic to Tumoral Lesions

**DOI:** 10.3390/biomedicines13040939

**Published:** 2025-04-11

**Authors:** Elizabeth Correa, Juan Pablo Rendón, Vanesa Bedoya-Betancur, Juliana Montoya, Julian Muñoz Duque, Tonny W. Naranjo

**Affiliations:** 1Medical and Experimental Mycology Group, CIB-UPB-UdeA-UDES, Corporación para Investigaciones Biológicas, Carrera 72 A # 78B-141, Medellin 050034, Colombia; ecorrea84@gmail.com (E.C.); juanren28@hotmail.com (J.P.R.); vbedoya24@gmail.com (V.B.-B.); julianamontoya9923@gmail.com (J.M.); 2Pathobiology Research Group QUIRON, Faculty of Agricultural Sciences, Universidad de Antioquia, Calle 70 # 52-21, Medellin 050036, Colombia; david.munoz@udea.edu.co; 3School of Health Sciences, Universidad Pontificia Bolivariana, Calle 78 B 72 A-109, Medellin 050034, Colombia

**Keywords:** model animal, colorectal cancer, BALB/c mice, preneoplastic lesions

## Abstract

**Background/Objetives:** This study standardized a chemically induced colorectal cancer (CRC) model using azoxymethane (AOM) and dextran sodium sulfate (DSS) in BALB/c mice, replicating the progression from preneoplastic lesions to adenocarcinoma observed in human colorectal carcinogenesis. **Methods:** The CCR-AOM/DSS model was standardized in male and female BALB/c mice. Two protocols were tested. Subsequently, the positive control group was established with nine evaluation points. Tumor progression was characterized histopathologically and corroborated by methylene blue staining and scanning electron microscopy. **Results:** Two cycles of 2% DSS combined with a single injection of AOM (10 mg/kg) were necessary to induce adenocarcinoma in 100% of the mice, with no significant sex-based differences in tumor development. Females showed earlier tumor susceptibility under certain protocols. Inflammatory processes played a critical role in tumorigenesis, with neutrophil infiltration and fibrosis observed. **Conclusions**: The findings align with previous reports, emphasizing the influence of DSS cycles, molecular weight, and mouse strain on model outcomes. This standardized model provides a reliable platform for the preclinical evaluation of novel preventive and therapeutic strategies for CRC.

## 1. Introduction

Colorectal cancer (CRC), that is, cancer affecting the colon and rectum, has some of the highest incidence and mortality rates, accounting for 10% of all incidences (1.9 million cases) and 9.4% of all deaths (more than 900,000 deaths) in 2020 among all types of cancers worldwide [1,2]. CCR carcinogenesis is associated with multiple risk factors, including tobacco and alcohol use, red and processed meat consumption, a high body mass index, physical inactivity, and chronic intestinal inflammation [3]. These factors contribute to the accumulation of genetic and epigenetic mutations in oncogenes or tumor suppressor genes, triggering the adenoma–carcinoma sequence [4]. This process occurs in three stages: (1) initiation, characterized by genetic damage in epithelial cells, primarily affecting the APC gene; (2) promotion, marked by excessive cell proliferation and abnormal colonic tissue growth, leading to the formation of premalignant lesions such as aberrant crypts (ACs), aberrant crypt foci (ACF), microadenomas (MAs), and adenomas; (3) progression, where these lesions acquire malignant and metastatic potential, ultimately developing into adenocarcinoma [5]. Notably, this transformation is slow, taking up to 10–15 years in humans, providing a broad window for early diagnosis and treatment [4].

Several preclinical studies have been conducted using both in vitro and in vivo models to understand the biology of CRC and to evaluate the implementation of prevention strategies, search for new therapeutic options, or both, with the in vivo models providing a paradigm for gaining insight into the context of the tumor microenvironment, compared with the in vitro models [6,7]. Most in vivo models used to date to study cancer as well as other diseases have focused on mice, likely due to the high genetic and biological similarity between mice and humans, thereby ensuring the high translational potential of the results obtained in such studies [8].

Several CRC animal models are currently available, including genetically engineered mice, tumor implantation, and chemical induction models [9]. The chemical induction models are characterized by the use of different chemical agents for tumor induction in animals, including some with the potential to cause DNA damage, such as azoxymethane (AOM), a widely used procarcinogen that is administered alone or combined with dextran sulfate sodium (DSS) and reportedly causes colitis and accelerates carcinogenesis [10].

This model has gained significant importance in recent years as it replicates certain pathological, histological, and molecular characteristics of the adenoma–carcinoma sequence observed in humans [9,10]. Although the mutations occurring in the colonic epithelium of mice may differ, studies indicate that Kras gene mutations are found at a low frequency (0–10%), and the P53 gene remains unmutated in this model. However, β-Catenin mutations have been reported, which occur downstream of the Wnt-Apc signaling pathway [11]. Furthermore, considering that the AOM/DSS CRC model is associated with an underlying inflammatory process, it has been reported that it leads to increased NF-κB transcription factor activity, elevated levels of proinflammatory cytokines such as TNF-α and IL-6, the enhanced infiltration of immune cells including macrophages, lymphocytes, and plasma cells, and the upregulation of the JAK/STAT3 signaling pathway [10].

Chemical induction protocols for AOM/DSS vary greatly with respect to parameters such as the AOM dose, the number of DSS cycles, and DSS concentration. In addition, variability has been observed in terms of the strain, sex, and age of the mice used in this model among the different studies. These are important aspects to consider, since some, such as sex, reportedly generate inconsistencies in the results as they can influence the susceptibility or resistance to CRC development [12].

Therefore, in this study, we aimed to standardize a mouse model of colitis-associated CRC chemically induced with AOM/DSS. The standardization of this model should facilitate the subsequent assessment of prevention strategies, treatments, or both for CRC, producing results that can support clinical studies, as the evolution of colorectal carcinogenesis observed is very similar to that in humans.

## 2. Materials and Methods

### 2.1. Animals and Housing Conditions

Male and female BALB/c mice between 8 and 10 weeks of age weighing 16–24 g were used in this study, and the mice were housed in the Animal Facility of the Corporation for Biological Research under controlled conditions (12 h light/dark cycle, 18–21 °C, 20 air changes per hour, and 45–70% relative humidity) and maintained in closed and ventilated cages containing wood wool and shavings bedding; sterile water and an autoclaved rodent lab diet were provided ad libitum. The cages, bedding, food, and water bottles were changed a minimum of once a week.

### 2.2. Experimental Design

#### 2.2.1. Pilot Study Experimental Design

In pursuit of determining the most effective cancer induction protocol, a preliminary investigation was carried out. AMO (Sigma-Aldrich, USA) and DSS (MP Biomedicals, USA) were used. This pilot study compared the tumoral development over 20 weeks in male and female mice, using four CCR induction protocols: P1 (one DSS cycle + AOM), CP1 (one DSS cycle without AOM), P2 (two DSS cycles + AOM), and CP2 (two DSS cycles without AOM). As shown in Figure 1a, 10 experimental groups (*n*: 12) were evaluated:
G1: Males treated with P1 (MP1).G2: Males treated with CP1 (MCP1).G3: Males treated with P2 (MP2).G4: Males treated with CP2 (MCP2).G5: Females treated with P1 (FP1).G6: Females treated with CP1 (FCP1).G7: Females treated with P2 (FP2).G8: Females treated with CP2 (FCP2).G9: Males without CCR induction (MNC).G10: Females without CCR induction (FNC).

Mouse weights were recorded every two weeks. Additionally, at evaluation intervals (weeks 5, 10, 15, and 20), three mice per group were sacrificed to assess carcinogenic progression through macroscopic and histological analysis.

#### 2.2.2. Positive Control Model Experimental Design

Upon obtaining the results from the pilot study, protocol 2 (intraperitoneal injection of 10 mg/kg of AOM at week 0, followed by two cycles of 2.0% DSS in drinking water at weeks 2 and 5) was selected as the optimal method for inducing colorectal cancer in BALB/c mice. This choice was made because, irrespective of gender (male or female), protocol 2 demonstrated complete colon tumor development within 15 weeks from the initiation of induction; Figure 2a. With this result, 32 new animals allocated to the positive control (PC; *n* = 24) and negative control (NC; *n* = 8) were used for the histopathological evaluation of tumoral progression for a period of 12 weeks; three animals in PC and one animal in NC were sacrificed at each evaluation point—weeks 1, 2, 3, 4, 5, 6, 8, 12 post initial cancer induction. PC mice received an i.p. injection of 10 mg/kg of AOM at week 0 and 2.0% DSS in drinking water at weeks 2 and 5 for 7 consecutive days. NC mice did not receive any AOM or DSS; Figure 2a.

### 2.3. Euthanasia and Colon Dissection

Following euthanasia by CO_2_ inhalation, the animals were positioned on an autopsy board, the legs were secured with pins, and the fur was cleaned with 70% ethanol. The skin and muscle layer were cut using sterile scissors, and the colon was removed from the cecum to the rectum, subsequently opened longitudinally and washed with phosphate-buffered saline (PBS) (Sigma-Aldrich, St. Louis, MO, USA) to remove the feces. The colon length and weight were recorded. Finally, 6 cm fragments were cut from the distal to the proximal part for lesion evaluation. The body weight of each animal was recorded twice weekly, and humane endpoints were defined as a weight loss of more than 25% or the development of rectal prolapse.

### 2.4. Evaluation of Preneoplastic Lesion Development

Preneoplastic lesions were detected using methylene blue staining, and the tissue was fixed with 10% buffered formalin for 24 h and then stained with methylene blue (Sigma Aldrich-USA) (0.2% [*v/v*] in PBS) for 30 min. The morphology of the crypts was then examined under a microscope at 4X and 10X magnification, and microphotographs were obtained of the most representative lesions. ACs were easily identified since they stain more intensely with methylene blue compared with normal crypts, have a larger diameter, are often found with oval lumens, and the epithelium generally appears thickened [13]. These ACs clump together to form ACF, which in turn proliferates by the fission of the crypts to form microadenomas (MAs) with more than 10 crypts, each less than 1 mm in size. The resulting MAs increase in size until the accumulation of additional genetic aberrations results in macroscopic adenomas [11]. Additionally, portions of colonic tissue obtained from some mice with three weeks of CRC induction were divided into 5 mm fragments, fixed with 2.5% gluraldehyde and 1% osmium tetraoxide, dehydrated with alcohol, and dried using a critical point dryer for their visualization by scanning electron microscopy (SEM) on the Apreo 2 SEM Thermo Scientific™ (Waltham, MA, USA) microscope at the Universidad Pontificia Bolivarina of Medellín.

### 2.5. Histopathological Analysis

The colonic tissues were sent to the Laboratory of Animal Pathology of the Faculty of Agricultural Sciences of the University of Antioquia (Universidad de Antioquia—UdeA) and were evaluated by a veterinary pathologist. Each tissue sample was composed of the most distal 6 cm of the colon of each animal and was cut into three sections (proximal, middle, and distal). The samples were embedded in paraffin wax and sectioned at 4 μm thickness. Each of the anatomical segments was stained with Hematoxylin and Eosin (HE). The slides were evaluated using an optical microscope Olympus CX43 (Olympus Corporation, Tokyo, Japan).

Histopathological analysis was performed by assigning a score from 1 to 6 (1, no lesion or condition; 2, mild; 3, mild to moderate; 4, moderate; 5, moderate to severe; 6, severe) to 31 parameters, including epithelial and glandular atrophy, goblet cell loss, epithelial and glandular hyperplasia, mitosis, aberrant crypts, epithelial and glandular dysplasia, neoplastic proliferation, neoplastic infiltration, GALT hyperplasia, erosion, ulceration, pigment, minerals, necrosis, apoptosis, congestion, edema, hemorrhage, thrombi, fibrin, exocytosis, neutrophils, eosinophils, macrophages, lymphocytes, plasma cells, fibrosis, and biological agents. Moreover, the report provides a final diagnosis.

### 2.6. Statistical Analysis

The results were expressed as means ± standard deviation (SD). A normality test was performed, and the statistical differences among experimental groups were evaluated each week. *p* values ≤ 0.05 were considered statistically significant. The results were analyzed and plotted using the GraphPad Prism 10.4 statistical package (GraphPad Software, San Diego, CA, USA).

## 3. Results

### 3.1. Pilot Study

During the development of the pilot study, the continuous monitoring of the animals’ health was conducted by the veterinarian, who recorded each animal’s body weight at least twice a week; animals from all groups that received DSS showed a weight decrease of approximately 5–24% during and after the DSS cycle; however, at least 5 days after the ends of DSS, all animals begin to recover their body weight and continued increasing it over time No significant weight loss was detected in FNC and MNC mice.

Regarding colon weight and length, the MP2 and FP2 groups had significantly heavier colons (*p* < 0.05) at weeks 15 and 20 after cancer induction compared with those in the MP1, FP1, and negative control groups at the same weeks; conversely, colons from the MP2 and FP2 groups at weeks 15 and 20 after cancer induction were significantly shorter than those of the negative control groups, with mean lengths of 10 and 13 cm, respectively. When the colons were observed macroscopically, most animals that received protocol 2, independently of gender male or female, had more intestinal masses at weeks 15 and 20 corresponding to tumors than those that received protocol 1, Figure 1b,c; some animals scheduled for euthanasia at week 20 of protocol 2 had to be euthanized before that endpoint due to rectal prolapse resulting from the tumors that caused intestinal obstruction, even before the planned time. Animals in the negative control and colitis control showed no tumors or macroscopic abnormalities.

With respect to the results of histopathological analysis, the middle and distal fragments of colonic tissue provided more information than the proximal fragment since these zones were more severely affected, as evidenced by the increased number and size of the tumors. According to macroscopic results, histopathologic analysis showed that those animals, both males and females, assigned to protocol 2 were diagnosed with adenocarcinoma by week 15, with scores of 3–4 according to the proliferation and neoplastic infiltration criteria; Figure 2d–f. In contrast, no signs of neoplasia or malignancy were consistently observed in the tissues of males and some females assigned to protocol 1, despite them showing a degree of neutrophil infiltration that indicated an inflammation process.

The histopathological results of both males and females in the negative control groups (MNC and FNC) showed no lesions or significant signs of abnormality at any time point, which is suggestive of healthy colonic tissues in these mice. As for the colitis control groups (FCP1, FCP2, MCP1, and MCP2), diagnoses of chronic colitis with signs of inflammation were obtained in most cases as early as week 5 after cancer induction.

These findings demonstrate that the administration of AOM combined with two cycles of DSS is required for the development of colorectal neoplasia, leading to a cancer diagnosis in both male and female mice. Considering these results, the scores of all histopathological parameters mentioned in the methodology were statistically analyzed to determine the optimal characteristics (sex and protocol) for the model to be used in subsequent studies. Some data did not present a normal distribution and were therefore analyzed using nonparametric tests. Table 1 shows the results of the Wilcoxon test that compared the results for the distal portion of the colonic tissue samples after week 15.

Since no significant histopathological differences were observed between tissue samples from males and females receiving two cycles of DSS (FP2 and MP2) at week 15 after cancer induction, the model to use in future studies consists of both female and male mice treated with an i.p. injection of 10 mg/kg of AOM at week 0 and 2.0% DSS in drinking water at weeks 2 and 5 for 7 consecutive days; mice were monitored for 16 weeks, allowing them to reach the final evaluation point before developing rectal prolapse.

### 3.2. Positive Control Group or CRC Model

The model and positive control group included both males and females for 16 weeks, with the administration of AOM and two cycles of DSS. The animals lost weight during the cycles of DSS and in the days following, owing to the clinical symptoms of colitis caused by the DSS treatment (Figure 2). Animals scheduled for euthanasia at week 16 were euthanized earlier because of rectal prolapse, and the analysis of this group was only conducted until week 12. Regarding the size and weight of colons, those tissues extracted from positive control mice showed a tendency to be shorter and heavier than those extracted from negative control mice, with some evaluation times having a clear statistical difference; Figure 2c,d. These results are related to the shortening of the colon due to an inflammation response in the first weeks of CRC induction and neoplastic involvement at week 12.

#### 3.2.1. Preneoplastic Lesions—Positive Control Group or Model

Methylene blue staining was used to determine the time of appearance and frequency of preneoplastic lesions during the development and evolution of adenocarcinoma. No preneoplastic lesions were observed in the intestinal tissue at week 1 after cancer induction; that is, no abnormalities were observed, like in the negative control groups. The first preneoplastic lesions (ACs and ACF) were detected at week 3 and to a lesser extent at week 4. MAs started developing from weeks 4 to 6, which evolved into lesions with a higher degree of malignancy, such as adenomas. The adenomas increased in both number and size over time, eventually progressing to polyps or tumors by week 8 after cancer induction (Figure 2e,f). Notably, detecting preneoplastic lesions was not possible at weeks 2 and 5 since the colonic tissue was highly irritated after the DSS cycles, making it difficult to obtain an accurate reading. These results show that the AOM/DSS-induced CRC model promotes a sequence of lesions that represent the entire carcinogenic process, starting from a normal colonic mucosa and evolving to preneoplastic lesions with different degrees of complexity and, finally, to adenocarcinoma.

Figure 3 and Figure 4 show microphotographs of the distal colon of mice with three weeks of evolution, taken by SEM (Thermo Scientific™, Waltham, MA, USA), in which both the healthy intestinal mucosa and preneoplastic lesions can be observed at different magnifications. In healthy mucosa, regular crypt lumens of similar sizes are observed, with the presence of goblet cells. In Figure 3D, apparently healthy epithelial cells are seen at 5000X magnification. In contrast, in Figure 4, preneoplastic lesions, FCAs (Figure 4A,B), a microadenoma (Figure 4E), and their respective enlargements indicated by an arrow (Figure 4C,D,F) are observed. In the right part of Figure 4A, an unchanged area is observed, apparently healthy, with uniform crypts of equal size; in the left part, indicated by the arrow, an FCA can be seen, made up of three aberrant crypts. This FCA is increased in Figure 4B, where the crypts are observed with the lumen larger than normal and hyperplasia of the epithelial cells around the lumen that appear higher than the surrounding mucosa. In Figure 4C, another FCA is observed, with a higher magnification in Figure 4D; this FCA has four aberrant crypts, and it also has crypts with larger and more elongated lumens than normal crypts. A microadenoma can be seen in the left part of Figure 4E and enlarged in Figure 4F. In this lesion, greater hyperplasia of the cells can be seen, causing the structure to protrude above the rest of the tissue that is apparently normal. It is evident that the cells that make up the lesion present dysplasia, with a morphology very different from that of the normal epithelium. On the other hand, in the tissue where these lesions occur, fewer goblet cells are observed, which is characteristic of abnormal tissue.

#### 3.2.2. Histopathology of Positive Control Group or CRC Model

The histopathological reports indicated that all animals were diagnosed with adenocarcinoma at week 12 after cancer induction, both in the distal and medial portions of the colon tissue samples. Moreover, moderate-to-severe neoplastic proliferation (grade 5 to 6) was observed in the colonic tissue of animals euthanized in the same week. These results confirmed the macroscopic findings, with up to 30 tumors identified in the colon of the animals, as previously described. Some animals were diagnosed with adenocarcinoma during the first weeks after cancer induction (week 3), presenting with histopathological lesions compatible with adenocarcinoma; this early result was not observed by macroscopic evaluation or methylene blue staining analysis. Figure 5a shows the severity of some of the histopathological characteristics associated with abnormal and accelerated growth (mitosis, ACs, dysplasia, and neoplastic proliferation and infiltration) in the distal portion of the colonic tissue of animals in the PC and NC groups. Animals in the PC group showed mild-to-moderate (grade 3) and moderate (grade 4) mitosis at all weeks, demonstrating high cell division rates from accelerated cell growth. Dysplasia was mild to moderate (grade 3) at week 2 after cancer induction, followed by neoplastic proliferation; the dysplasia severity decreased as the area of neoplastic tissue increased. Regarding the medial portion of the colon fragments, some signs of malignancy were observed at week 2 after cancer induction, with reports of mild-to-moderate neoplastic proliferation (grade 3) that increased over time to moderate to severe (grade 5) at week 12 after cancer induction. In addition, the abnormal tissue occupied nearly the entire mucosa, as seen in the microphotograph of the histological sections (Figure 5b).

Another evaluated histopathological parameter was the presence of neutrophils, which indicates inflammation. This is evident in Figure 5b in which neutrophil infiltration can be observed, mainly in the image from week 2 after cancer induction. The ulceration, possibly resulting from tissue damage caused by CCR induction with AOM/DSS and inflammation, exhibited low-to-moderate severity levels between week 4 and 6. As for fibrosis, which develops as a consequence of the accelerated repair of damaged and inflamed tissue, high severity levels (up to grade 5) were observed from the first DSS cycle (week 2 post-induction) to the final evaluation moment (week 12).

## 4. Discussion

This study described the standardization of an animal model of CRC chemically induced with carcinogenic AOM and the promoter, DSS, in BALB/c mice, replicating the sequence of preneoplastic lesions to adenoma and then to adenocarcinoma that typically occurs during colorectal carcinogenesis and is associated with the inflammatory process in humans. For this purpose, a pilot study was performed to evaluate the administration of one or two cycles of DSS, AOM, or neither, in both male and female mice.

The results suggest that combined AOM/DSS has high potential to lead to the development of CRC, as reported by several authors [11,14]. The mutagenic action of AOM on colonic cells and the irritant action of DSS in the colitis phase favor the development of carcinogenesis. In addition, the combined use of AOM and DSS was reportedly required for the development of adenocarcinoma, as previous studies found that 10 mg/kg of AOM alone was not sufficient for the development of colon polyps in mice 20 weeks after cancer induction. Similarly, DSS alone leads to chronic colitis, although not to the development of adenocarcinoma, as observed in the colitis control mice receiving protocols 1 (CFP1 and CMP1) and 2 (CFP2 and CMP2) and those previously reported [14,15]. We also confirmed that the number of DSS cycles is an important parameter for the development of neoplasia in BALB/c mice, since two 2% DSS cycles with a duration of 7 days each were required in our study to detect tumors macroscopically that were later histopathologically diagnosed as adenocarcinoma in 100% of the cases. This is consistent with most studies that reported the use of two or more DSS cycles with AOM for CRC induction [16,17].

Therefore, the number of DSS cycles depends on the mouse strain or genetics. BALB/c and C57/BL6 are among the most common mouse strains for CRC induction using the AOM/DSS system. BALB/c mice are immunosuppressed and considered to be more susceptible to tumor development. As previously published, BALB/c mice required one injection of AOM (10 mg/kg) with a shorter course (less than 7 days) of 1% DSS for 100% of the mice to develop tumors [16,18]. Conversely, C57/BL6 mice are immunocompetent and more resistant to tumor development, and, therefore, many studies have administered AOM in combination with up to four cycles of DSS or with a concentration higher than 2% to induce CRC in these mice [19]. Given these characteristics, the BALB/c strain ensures tumor generation at a lower concentration and with fewer DSS cycles. For these reasons, this strain was selected in the present study to standardize the chemically induced CRC model.

Notably, DSS molecules must be 36,000 to 50,000 Da in size to cause colitis in mice [20], since previous experiments used this same reagent but with a larger size (500,000 Da), and no inflammation or irritation was observed in the colon. Moreover, no tumors developed in the colons of mice when the reagent was administered together with AOM [20,21].

In our model, no statistically significant differences were observed in colorectal adenocarcinoma development between male and female mice when two cycles of 2% DSS and a single AOM injection (10 mg/kg) were used. However, results from the pilot study suggested that females exhibited greater susceptibility to the AOM/DSS system. This was evidenced by the presence of tumors and histopathological diagnoses of adenocarcinoma at week 15 post-induction in females treated with protocol 1 (one DSS cycle + AOM). In contrast, males under the same protocol did not develop intestinal tumors, and histopathological analysis only revealed colitis. These findings differ from previous reports, as some studies suggest sex-based differences in CRC development. Estradiol has been proposed as a protective factor, as it modulates inflammation by increasing the expression of Nrf2 (nuclear factor erythroid 2) [12], downregulates PD-L1 (programmed death-ligand 1), potentially altering the immunosuppressive tumor microenvironment, and favorably influences microbiota composition [22]. Conversely, androgens have been associated with an increased risk of CRC as they have been shown to promote tumor growth and increase the number of colorectal tumors [12,23,24]. These observations are based on both epidemiological studies in humans and experimental research in animal models. However, conflicting results have also been reported [25,26], indicating that estradiol does not always exert a protective effect and may, under certain conditions, promote carcinogenesis. This has been observed in advanced tumor stages or when the tumor microenvironment is in a hypoxic state [22], as well as in male mice undergoing orchiectomy and subsequent estradiol replacement therapy [24]. The increased susceptibility of females to develop CRC with protocol 1 observed in this study may be attributed to a combination of hormonal, immune, microbial, and even genetic factors. Further research is necessary to fully elucidate the underlying mechanisms driving these differences.

Regarding the global epidemiology of CRC, this neoplasm reportedly occurs more frequently in men than in women, with approximately 400,000 more cases in men than in women in 2020, according to GLOBOCAN (2). However, this trend differed in Colombia, as 5823 cases were reported in women and 4989 in men in 2020, according to the same organization [27], indicating that the proposed hypothesis is not always met. Therefore, obtaining preclinical experimental conclusions is important, whether preventive, diagnostic, or therapeutic, in both males and females, to shed light on the context of the disease in these two populations.

Another important aspect to consider for the CRC model with combined AOM/DSS is the age of the mice at the start of the chemical induction. Ideally, older mice are used to represent or simulate the age at which CRC occurs most frequently in humans, that is, 60 years of age and older. In our pilot study, 8-week-old males and 9-week-old females were used, since females are generally smaller. This age was selected because the animals are adults, with all their systems having reached maturity, and generally weigh 18–22 g, which is most frequently reported in the literature for animal models that use this chemical induction system.

Clinically, body weight monitoring showed that mice in the pilot study behaved as reported in previous studies, since they lost 5 to 25% of their body weight during and after DSS treatment, which is explained by the symptoms (diarrhea, dehydration, and gastrointestinal bleeding) associated with colitis induced by the DSS [14,15,19]. Therefore, a rehydration stage was included after each DSS cycle in the CRC induction protocol to improve the health of the animals, thereby facilitating their successful recovery and ensuring that they could reach the scheduled endpoint [21].

Regarding the weight and size of the colon dissected during the pilot study, significant differences were only observed at some time points for the tissue samples from animals receiving protocol 2. Tissues with greater neoplastic involvement tended to be heavier and smaller than those of the negative controls. This finding is similar in humans in cases of neoplasia or inflammatory bowel disease (IBD) since the intestine loses elasticity and decreases in size because of the increase in collagen fibers from fibrosis [17,28].

Based on the macroscopic and histopathological analysis of the colonic tissue samples, the appropriate conditions were established according to the evaluated parameters to develop a CRC induction model with the AOM/DSS system, with at least two DSS cycles required to generate tumors or adenocarcinoma in all female and male mice.

It was confirmed that methylene blue staining is a useful technique to recognize the morphological characteristics of preneoplastic lesions. Measuring preneoplastic lesions provided insight into the sequence of the adenocarcinoma development and proved that the chemically induced CRC involves a series of epithelial changes in BALB/c mice, leading to the transformation of healthy colonic tissue to cancerous tissue. ACs initially appear in the colonic epithelium and are characterized by abnormally thickened epithelial cells, larger size, and more intense methylene blue staining compared with those in normal crypts. These ACs clump together and become ACF, which develop into MAs as the cells continue to grow. MAs then evolve into adenomas, which are considered benign and reversible, although they can acquire cancerous characteristics and become malignant tumors. The times in which these lesions occurred are comparable to those published by De Robertis et al. in 2011, since the first aberrant crypts and FCA are reported in week 3 with the CRC induction system—AOM/DSS in BALB/c mice [11]. Likewise, the images taken at high magnification using SEM allowed us to corroborate the morphology previously visualized at low magnification, which demonstrated that classic elevated lesions are present in our model and that their appearance is similar to those found in other studies; for example, Paulsen et al. induced colorectal cancer in F344 rats by injecting 1,2-dimethylhydrazine, and after 16 weeks, they observed elevated FCA due to hyperplasia and dysplasia [29].

Microscopic, progressive changes associated with inflammation were observed, followed by tumorigenesis. During episodes of colitis, the tissue incurred significant damage, including the loss of epithelial organization and goblet cells and extensive neutrophil infiltration in the epithelium, lamina propria, and submucosa, which is indicative of active inflammation and known as cryptitis and crypt abscesses. Collagen fibers also appeared, indicating fibrosis apparently due to the accelerated response to inflammation. However, after the administration of DSS, epithelial regeneration apparently occurred, and goblet cells were again observed in the colonic epithelium at week 3, although with dysplastic morphology and foci of neoplastic proliferation. Subsequently, medium grades of mucosal ulceration were observed at week 4, accompanied by increased proliferation and neoplastic infiltration. These results are like those previously described in CKD mice showing intestinal epithelial ulceration as a consequence of colitis caused by DSS and diagnoses of colonic neoplasm at week 3 [30].

All the aforementioned changes are associated with genomic damage and colitis induced by the combination of AOM and DSS and are comparable to what occurs in patients with IBD with either ulcerative colitis or Crohn’s disease [21]. Moreover, moderate inflammation occurred in the model proposed in this study and was an underlying factor, although it was expected since the model is based on the induction of colitis as the promoter of carcinogenesis. In this regard, Zheng et al. (2022) demonstrated through an extensive review (although some authors refer to the protective factor of neutrophils with regard to carcinogenesis) that chronic neutrophil infiltration in the tissue could be involved in promoting tumor progression by promoting cancer cell proliferation, invasion, angiogenesis, and immunosuppression, suggesting that neutrophil-targeted immunotherapy could be a promising treatment for CRC [31,32]. Infiltration into other cells (lymphocytes, eosinophils, and macrophages) was not significant compared with that in the negative controls at the different time points. However, the tumor microenvironment associated with the inflammatory response is crucial for the development of carcinogenesis [33].

Finally, this study demonstrates that combined AOM/DSS strongly and rapidly promotes the development of inflammation-associated carcinogenesis [11,34]. We observed foci of adenocarcinoma in the early stages of cancer induction (week 3), as well as the development of macroscopically visible tumors at week 8, with moderate grades of neoplastic infiltration 2 to 3. In contrast, other studies reported that it can take up to 12 weeks after cancer induction to develop a colonic neoplasm in BALB/c mice [35] and up to 10–20 weeks to develop adenocarcinoma with mild grades of neoplastic infiltration in other mouse strains, such as CKD [33].

Unlike traditional models that typically use only one sex, our model is standardized and characterized for both male and female subjects, allowing for more comprehensive, sex-specific insights into CRC development and progression. This enhances the generalizability of results to the broader population. Additionally, by providing detailed characterization, the model offers a more robust platform for evaluating new therapies or modifications to conventional treatments for CRC, making preclinical research more relevant and applicable. Furthermore, this is the first AOM/DSS-based CRC model developed in our region, establishing an accessible platform for researchers and institutions that require a reliable model for preclinical testing.

## Figures and Tables

**Figure 1 biomedicines-13-00939-f001:**
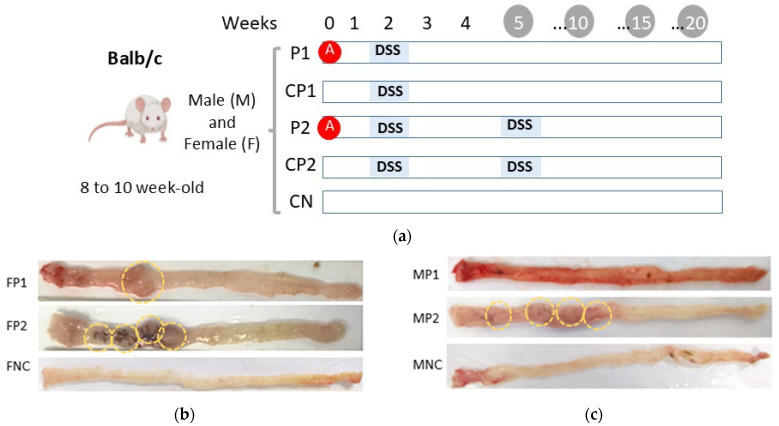
The pilot study. (**a**). Experimental design. Mice received 10 mg/kg of azoxymethane (AOM) (red circle) intraperitoneally and 2% dextran sulfate sodium (DSS) (blue bar) in drinking water. Two protocols were tested in both males (M) and females (F): P1 (one cycle of DSS) and P2 (two cycles of DSS). FP1 and MP1 and FP2 and MP2 model groups received AOM and one or two cycles of DSS, respectively. The colitis control groups, CFP1 and CFP2, as well as CMP1 and CMP2, received only one or two cycles of DSS, respectively. The FNC and MNC groups received no treatment. Weeks 5, 10, 15, and 20 were the time points for euthanasia and subsequent analysis (gray circle). (**b**,**c**) The colons of females and males at week 15 after cancer induction. The yellow dashed lines indicate the tumors. (**d**–**g**) The severity of the histopathological characteristics of the distal fragment of the colonic tissue samples for female and male mice in weeks 10 and 15, respectively. ** *p* < 0.005, and *** *p* < 0.0005.

**Figure 2 biomedicines-13-00939-f002:**
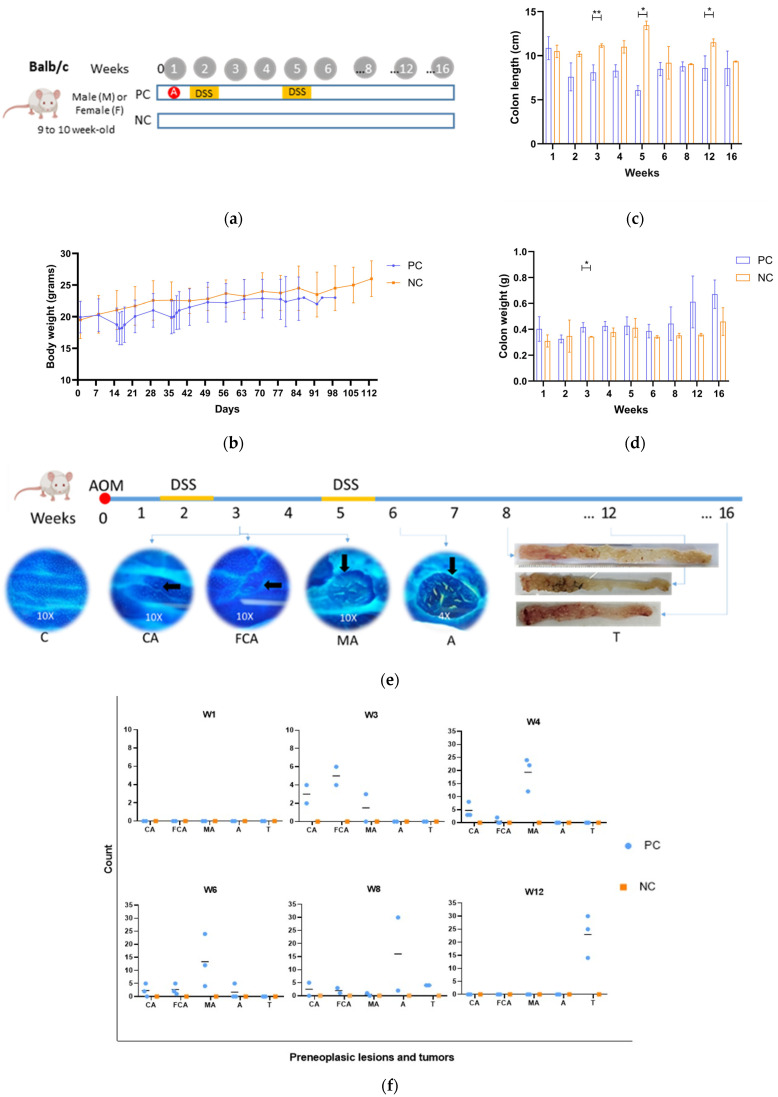
Positive control (**a**). Positive control model experimental design. Mice were administered 10 mg/kg of azoxymethane (AOM) (red circle) intraperitoneally and received two cycles of 2% dextran sulfate sodium (DSS) (yellow bar) in drinking water. Weeks 1, 2, 3, 4, 5, 6, 8, 12, and 16 were the time points for euthanasia and analysis (gray circle). PC (positive control group) or model and NC (negative control group). (**b**). Body weight in grams of the mice during the study period. The mean value of six mice from each PC group and two mice from the NC group is shown for each week. (**c**): Colon lengths. (**d**). Colon weights. (**e**): Time of appearance of the preneoplastic lesions or tumors during the development of the CRC mice model. (**f**). Number of lesions or tumors per week. W1 to W12: weeks 1 to 12. C, normal crypt; CA, aberrant crypt; FCA, aberrant crypt foci; MA, microadenoma; A, adenoma; T, tumor. The arrows point to the lesions. * *p* < 0.05, ** *p* < 0.005.

**Figure 3 biomedicines-13-00939-f003:**
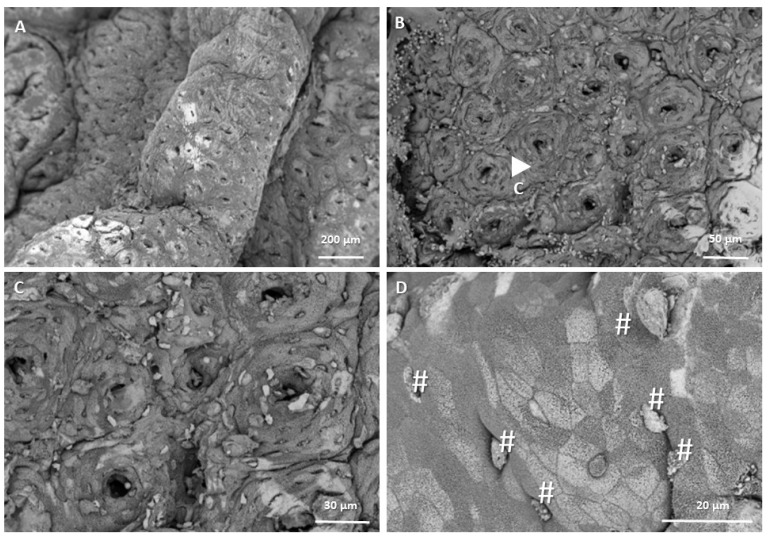
Microphotographs of the healthy colonic mucosa, acquired by SEM with a T1 detector at different magnifications: (**A**): 250X, (**B**): 1000X, (**C**): 2000X, and (**D**): 5000X. The white arrow in microphotograph (**B**) indicates the area magnified in (**C**). # in microphotograph (**D**) indicate the goblet cells.

**Figure 4 biomedicines-13-00939-f004:**
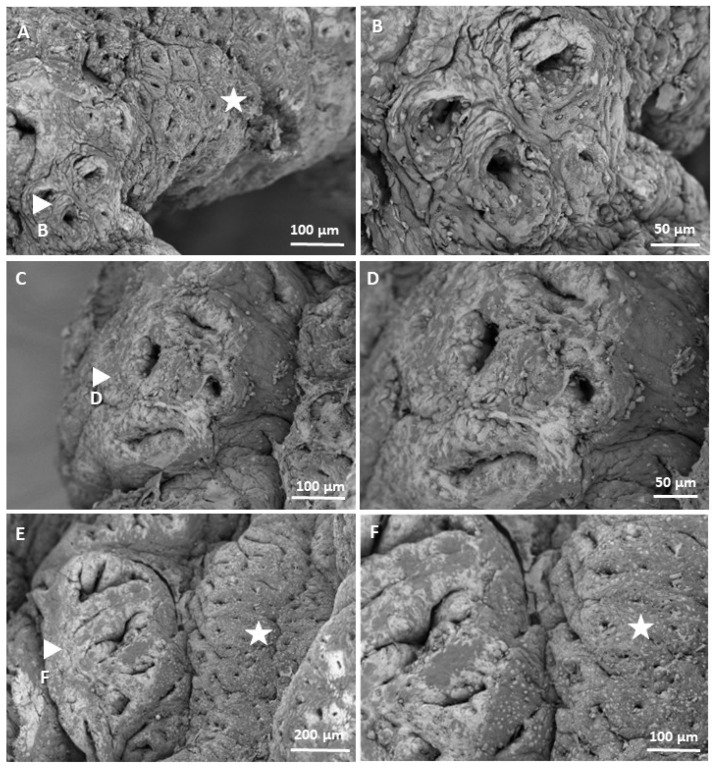
Microphotographs of the colonic mucosa with preneoplastic lesions, acquired by SEM with a T1 detector at different magnifications: (**A**): 500X, (**B**): 1200X, (**C**,**F**): 650X, (**D**): 1000X, and (**E**): 350X. (**A**–**D**): foci of aberrant crypts. (**E**,**F**): microadenoma. The arrows in (**A**,**C**,**E**) indicate the areas that are enlarged in (**B**,**D**,**F**), respectively. The stars indicate healthy areas with normal crypts.

**Figure 5 biomedicines-13-00939-f005:**
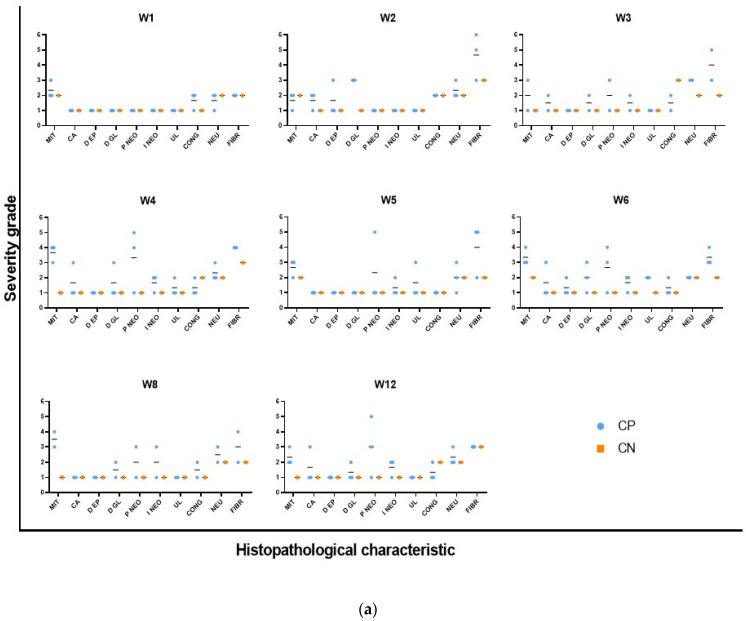
(**a**) Histopathological characteristics of the control groups. Positive control (PC) and negative control (NC) groups. Weeks 1 to 16 (W1 to W12). MIT, mitosis; CA, aberrant crypt; EPD, epithelial dysplasia; GLD, glandular dysplasia; NEOP, neoplastic proliferation; NEOI, neoplastic infiltration; UL, ulceration; CON, congestion; NEU, neutrophils; FIB, fibrosis. (**b**) Representative images of the histopathological analysis of the colon. No significant lesions were observed in the CN group. Note the presence of small neoplastic lesions from PCW3 progressing to extensive masses up to PCW12. Magnifications: NC and W2: 40X; W3, W4, and W5: 10X; W8 and W12: 4X.

**Table 1 biomedicines-13-00939-t001:** Wilcoxon test results for histopathological findings.

Experimental Groups Compared	Wilcoxon Test Result	*p*-Value
FP1 × FP2	Sig ***	0.0003
MP1 × MP2	Sig **	0.0020
FP2 × MP2	NS	0.9355

All animals were included in the model groups treated with AOM and DSS. FP1 and MP1, females and males, respectively, for protocol 1 (one DSS cycle); FP2 and MP2, females and males, respectively, for protocol 2 (two DSS cycles); Sig, significant differences; NS, no significant differences. ** *p* < 0.005; *** *p* < 0.0005.

## Data Availability

The original contributions presented in this study are included in the article. Further inquiries can be directed to the corresponding author.

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
