# Peer review of "Standardization of a Preclinical Colon Cancer Model in Male and Female BALB/c Mice: Macroscopic and Microscopic Characterization from Pre-Neoplastic to Tumoral Lesions"

_biomedicines, 2025, doi:10.3390/biomedicines13040939_

Round 1
Reviewer 1 Report
Comments and Suggestions for Authors
The manuscript presents a well-designed study to standardize a chemically induced colorectal cancer (CRC) model using AOM/DSS in BALB/c mice. The experimental approach is rigorous, and the data are robust, demonstrating the utility of this model for preclinical research. However, revisions are required to enhance clarity, address methodological ambiguities, and strengthen the discussion.
Major Comments
Introduction: The description of the "adenoma-carcinoma sequence" (Page 1) overlaps with background information. Condense this section to avoid redundancy.
Methods: The pilot study design (Section 2.2.1) is overly complex. Use a table to summarize experimental groups (e.g., treatments, sample sizes, endpoints).
Terminology: Avoid informal phrasing (e.g., "data no shown or Supporting" → "Data not shown").
Key Findings: The demonstration of 100% adenocarcinoma induction with two DSS cycles is compelling. However:
- Sex Differences: The discussion of earlier tumor susceptibility in females (Page 12) lacks mechanistic exploration. Cite relevant studies and propose hypotheses (e.g., hormonal influences).
- DSS Molecular Weight: The critical role of DSS molecular weight (36,000–50,000 Da) is mentioned but unsupported by citations. Add references.
- Innovation: Highlight how this model improves upon existing AOM/DSS protocols.
Author Response
Reviewer 1
We sincerely appreciate the editor and reviewers for their time and valuable feedback on our work. We have carefully addressed all of their comments, which have greatly improved the clarity and presentation of our study. Below, we provide our responses to each of the reviewers' comments. In the revised manuscript, all specific changes have been highlighted in yellow for easy reference.
Comment 1: Introduction: The description of the "adenoma-carcinoma sequence" (Page 1) overlaps with background information. Condense this section to avoid redundancy.
Response 1: We appreciate the reviewer’s suggestion and we have removed overlapping background information. This adjustment improves clarity and ensures that the introduction remains focused and concise.
Comment 2: Methods: The pilot study design (Section 2.2.1) is overly complex. Use a table to summarize experimental groups (e.g., treatments, sample sizes, endpoints).
Response 2: For better clarity, Figure 1(a) has been adjusted to specify the groups, the differences between protocols 1 and 2, the euthanasia time points, the number of animals, etc. Additionally, section 2.2.1 has been adjusted to improve clarity.
Comment 3: Terminology: Avoid informal phrasing (e.g., "data no shown or Supporting" → "Data not shown").
Response 3: It has been adjusted in the text according to the recommendation
Comment 4: Key Findings: The demonstration of 100% adenocarcinoma induction with two DSS cycles is compelling. However:
- Sex Differences: The discussion of earlier tumor susceptibility in females (Page 12) lacks mechanistic exploration. Cite relevant studies and propose hypotheses (e.g., hormonal influences).
Response: An analysis in this regard, supported by references, was included in the discussion (pages 13 and 14).
- DSS Molecular Weight: The critical role of DSS molecular weight (36,000–50,000 Da) is mentioned but unsupported by citations. Add references.
Response: Reference supporting that the molecular weight of DSS is essential for the development of colitis has been added. Page 12, reference 20 (Eichele DD, Kharbanda KK., 2017).
- Innovation: Highlight how this model improves upon existing AOM/DSS protocols.
Response: A paragraph was added at the end of the discussion to emphasize the innovative nature of this study and its relevance to our regional context.
Reviewer 2 Report
Comments and Suggestions for Authors
-The study has several strengths including comprehensive characterization (micro- and macroscopic), covering pre- and neoplastic lesions, and studying both genders.
-A few suggestions to the authors:
- Please justify the use of Balb/c as opposed to C57Bl/6 mice.
- What could be the molecular fingerprints of the different stages (e.g., Ki-67, p53... etc.)?
- Why did females show earlier susciptability than males?
- Did the authors consider doing some functional tests like colonic motility?
- Discussing the influence of the gut microbiome and immune mediators on colon cancer progression is recommended.
- The resolution of some of the figures must be improved.
- In the statistics section, please mention the normality tests used, their result (i.e. were the data normally distributed or skewed?), and what tests were used for comparisons. Also, please justify the use of SEM instead of SD.
- How did the authors decide on the number of mice used? Did the authors do power analysis of their sample size?
Author Response
We sincerely appreciate the editor and reviewers for their time and valuable feedback on our work. We have carefully addressed all of their comments, which have greatly improved the clarity and presentation of our study. Below, we provide our responses to each of the reviewers' comments. In the revised manuscript, all specific changes have been highlighted in blue easy reference.
Comment 1: Please justify the use of Balb/c as opposed to C57Bl/6 mice.
Response 1: In the discussion section, an explicit justification for the use of the BALB/c strain has been added.
Comment 2: What could be the molecular fingerprints of the different stages (e.g., Ki-67, p53... etc.)?
Response 2: We have added a paragraph in the introduction (page 2) that incorporates information regarding the molecular markers associated with the different stages of the murine model using the AOM/DSS system, including p53, Kras and others.
Comment 3: Why did females show earlier susceptibility than males?
Response 3: An analysis in this regard, supported by references, was included in the discussion (pages 13 and 14).
Comment 4: Did the authors consider doing some functional tests like colonic motility?
Response 4: Although colonic motility testing was not included in this study, we recognize its potential value and will consider incorporating it in future research to further enhance our understanding of the model.
Comment 5: Discussing the influence of the gut microbiome and immune mediators on colon cancer progression is recommended.
Response 5: Although these topics were not extensively discussed in the current manuscript, a brief discussion on the influence of immune mediators and the microbiota on CRC progression has been included in the introduction and discussion sections. Currently, our group is investigating the relationship between the microbiota, the adenocarcinomatous process, 5-FU treatment, and microbiota modulation using probiotics in the AOM/DSS murine CRC model. A separate publication is being prepared to present the relevant analyses.
Comment 6: The resolution of some of the figures must be improved.
Response 6: The images have been enhanced to improve their resolution and clarity
Comment 7: In the statistics section, please mention the normality tests used, their result (i.e. were the data normally distributed or skewed?), and what tests were used for comparisons. Also, please justify the use of SEM instead of SD.
Response 7: The data were analyzed using the Kolmogorov-Smirnov normality test, which, as mentioned in the manuscript (page 5), indicated that they did not follow a normal distribution. Therefore, the Wilcoxon test was used to compare the experimental groups across protocols and sexes. In section 2.6, a typographical error was identified. The correct measurement used was SD, which accurately reflects the variability or dispersion of the data.
Comment 8: How did the authors decide on the number of mice used? Did the authors do power analysis of their sample size?
Response 8: To calculate the number of animals for pilot study per group we use the standard formula for sample size calculation in comparative studies:
n=(2(σ2)(Zα/2​+Zβ​)2)/​ Δ2
Where:
- n = number of animals per group.
- Zα/2 = Z-value corresponding to the significance level (α). For α=0.05 (significance level of5%), Zα/2=1.96
- Zβ= Z-value corresponding to the desired power (1−β). For a power of 80% (1−β=0.80), Zβ=0.84.
- σ2 = estimated variance of the data. It is often assumed to be σ2=1.
- Δ=d×σ
- d = effect size, in our case, is 1.14
Thus, a sample size of n = 12 animals per experimental group was obtained, and euthanasia was performed at the time points specified in the article.
Round 2
Reviewer 2 Report
Comments and Suggestions for Authors
Comments satisfactorily addressed.